# The Influence of Buyer Power on Supply Chain Pricing with Downstream Competition

**Qiu Zhao**

School of Business Administration, Northeastern University, Shenyang 110169, China; zhq340621@163.com or 1510504@stu.neu.edu.cn

**Abstract:** This paper aims to investigate the impact of buyer power on the wholesale price and retail price of, in the case, downstream competition. Based on a summary of the competitive characteristics of China's retail market, a model of a vertical market was constructed to examine the influence of buyer power on the pricing decisions of manufacturers and retailers, and to analyze the mechanism of price decisions. The results showed that the buyer power of national retailers reduced the wholesale price, but the impact on local retailers remained uncertain. Although increasing buyer power initially increased the local retailer's wholesale price and caused the 'waterbed effect', we found that this effect reverted when the buyer power reached a point at which the 'anti-waterbed effect' appeared. The opposite was true of the retail price. However, buyer power reduced the average retail price, and consumer welfare improved.

**Keywords:** buyer power; retailer competition; waterbed effect; pricing strategy; national retailer; local retailer

## 1. Introduction

Market power, corporate behavior, and market performance have always been the core issues of economic research, and have also been the focus of antitrust regulation. The market power that economists are concerned with is the seller power relative to the buyer power (mainly consumers) [1]. However, another opposing market power has gradually attracted the attention of the academic community in recent decades [2–5]. This market power is the downstream buyer (mostly enterprises) relative to the upstream seller, which is called buyer power [6–8]. Buyer power is especially common in the retail industry. With the development and expansion of large-scale retail organizations such as Wal-Mart, Carrefour and Tesco, retailers with buyer power have become a common phenomenon. These retailers have ever increasing consumer resources, and upstream manufacturers increasingly rely on retailers. Retailers are able to put forward various requirements to manufacturers, regarding product quality and supply terms, by virtue of their buyer power. This behavior changes the original vertical relationship of the industrial chain, causing some vertical problems within.

As Biely et al. argue in their paper, with increasing market concentration, market power can have an impact on sustainability [1]. This paper investigates buyer power, which is also a kind of market power that flows upstream. For example, the market power of large supermarkets can affect the health of consumers [9]. This relates to sustainability (e.g., social sustainability related to food security). Large enterprises may affect the sustainability of an industry by altering their own business operations, such as the 'greening' of Walmart [10], and DuPont phasing out chlorofluorocarbons (CFS) [11], which illustrate the relevance of market power on environmental and social sustainability. Buyer power in the retail market can also trigger conflict between suppliers and retailers. In terms of China's retail market, the conflict between Gome and Gree in 2004, the conflict between RT-Mart and Blue Moon in 2015, and the conflict between China Resources Vanguard and Walch in 2015 are

salient examples [12–14]. These conflicts affect the sustainable development of the retail industry and suppliers, and the economic sustainability influenced by buyer power.

There are also concerns among academics and antitrust authorities regarding buyer power [15]. It is generally recognized by researchers that buyer power will reduce the wholesale price of large retailers who own buyer power, or bring other more favorable terms. However, in order to make up for this loss, upstream suppliers may increase the price for other retailers, meaning that buyer power may worsen the exchange provision of competitors. This is called the 'waterbed effect'. The 'waterbed effect' essentially reflects the influence of buyer power on competitors, but researchers have not reached a consensus. Some scholars believe that there is a 'waterbed effect', but others argue that buyer power will not worsen the trading conditions of competitors; on the contrary, it will bring preferential clauses to competitors, which is called the 'anti-waterbed effect'. Furthermore, there are some studies suggesting that buyer power does not affect competitor's prices at all; in other words, the 'waterbed effect' and the 'anti-waterbed effect' will not take shape. If buyer power has a waterbed effect, a powerful enterprise can crowd out competitors by raising the trading price for competitors, which will result in a change in the market structure. It is clear that buyer power could be unsustainable for suppliers and competitors.

In addition, Galbraith believed that buyer power would bring retailers preferential clauses and, at the same time, that consumers would receive benefits, such as decreased retail prices and improved social welfare. This is called the Galbraith Hypothesis [2]. Most researchers believe that the Galbraith hypothesis is indeed true under certain conditions because buyer power may reduce retail price. However, many other scholars doubt this hypothesis, arguing that the increase in buyer power will raise the retail price. According to Biely et al.'s criteria [1], from the perspective of retail price and consumer welfare, this is related to economic sustainability. Buyer power may also have an impact on supplier innovation incentive [16–20], which can have adverse effects on the environment or health of the consumer, this is related to environmental and social sustainability.

Why does buyer power lead to conflicts between the upstream and downstream? What effect does buyer power have on wholesale and retail prices in the market? What is the hidden mechanism? This paper analyzes the price effect of buyer power and the interaction between buyer power and market competition via the model of downstream competition, and upstream and downstream non-cooperative transactions. Xiao et al. used the non-cooperative transactions approach in his research, and considered the impact of the market environment on pricing decisions in the supply chain [21]. Intuitively, the more competitive the retail market, the weaker retailers' buyer power should be. Manufacturers' dependence on a single retailer has declined on account of the fierce competition in the retail market, resulting in weaker buyer power of retailers compared with manufacturers. In addition, the competitive environment of the retail market may affect the role of buyer power, which in turn may have an impact on the competitive environment of the retail market. For example, Crawford and Yurukoglu found in their empirical work on the cable TV market that downstream market competition played a role in buyer power [22].

The rest of the article is organized as follows. In Section 2, we introduce existing literature on buyer power and the waterbed effect, and buyer power and the Galbraith hypothesis. In Section 3, we briefly describe the characteristics of China's retail market, and a basic model is outlined. We then analyze the effects of buyer power on wholesale prices and retailer prices in Section 4. In Section 5, we explore the influence of market competition on the price effect of buyer power by numerical simulation. We outline our conclusions in Section 6. The proofs are in the Appendices A and B.

## 2. Literature Review

Robinson proposed the concept of monopsony power as representative of the market power of the buyer relative to the seller [23]. This concept was proposed by the analog monopoly power, reflecting the market power of downstream monopoly buyers relative to upstream sellers in a perfect competition environment. A clear conception of buyer power was put forward in 1952. Galbraith, an American economist, proposed the concept of 'countervailing power', which meant that the increase

in the market power of upstream enterprises related to downstream enterprises would bring about another, opposite market power [2]. Countervailing power is essentially a form of buyer power, meaning the power of downstream enterprises relative to upstream enterprises. The Organization for Economic Co-operation and Development (OECD) defined buyer power as "[...] the situation which exists when a firm or a group of firms, either because it has a dominant position as a purchaser of a product or a service or because it has strategic or leverage advantages as a result of its size or other characteristics, is able to obtain from a supplier more favorable terms than those available to other buyers" [24] (p. 10). Similarly, Dobson et al. and Clarke et al. defined buyer power as "[...] a firm or group of firms obtain from suppliers more favorable terms than those available to other buyers or would otherwise be expected under normal competitive conditions" [25,26] (p. 5, p. 8). Mills defined buyer power as "[...] the ability of large buyers to obtain preferential terms of sale from suppliers that are not available to small buyers" [27] (p. 66). Finally, Chen summarized these concepts, concluding that the meaning of buyer power is broader. It refers to the ability of retailers to obtain lower than the normal supply price or superior supply conditions. When the upstream market is completely competitive, the buyer power of the retail market is a monopsony power, and when the upstream is in a state of incomplete competition, the buyer power is countervailing power or bargaining power [19]. We will also adopt the concept proposed by Mills, viewing buyer power as the ability of large retailers to obtain preferential terms.

Previous literature on buyer power has mainly focused on two aspects of this concept, including the sources and influences of buyer power. Regarding the source of buyer power, Inderst and Wey believed that when the total profit function of an industry chain was concave, downstream enterprises could improve their buyer power [28,29]. Normann et al. supported this argument through experimental simulations [30]. From the perspective of the buyer's reverse acquisition capabilities, Katz, Fumagalli and Motta, and O'Brien explained the formative mechanism of buyer power [31–33]. Some scholars believe that consumer preference is an important source of buyer power. The product substitution generated in a one-stop procurement process could change the negotiation mode between manufacturers and retailers, which could affect the buyer power of retailers [34,35].

Regarding the analysis of buyer power and the waterbed effect. Inderst and Wey, and Inderst and Valletti, argued that the buyer power of large downstream retailers would increase the supply price of competitors, resulting in a waterbed effect [28,36]. Majer proved the existence of the waterbed effect and, further, found that the strength of the waterbed effect was related to the degree of competition in the downstream market [37]. King studied the conditions of the waterbed effect, and found that the waterbed effect is related to downstream competition, upstream cost characteristics, and the demand of the market [38]. However, Chen showed that an increase in buyer power reduced the wholesale price of marginal retailers, known as the anti-waterbed effect [4]. As buyer power reduced the profits of the manufacturer, the manufacturer aimed to reduce the wholesale price of marginal retailers to expand their market share to compensate for the loss of profits of the leading retailer. Erutku reached a similar conclusion to Chen in terms of the effects of strong buyer power [4,39]. All of these studies suggest that buyer power has an impact on the economic sustainability of the retail sector.

Regarding the analysis of buyer power and the Galbraith hypothesis, in essence, the Galbraith hypothesis reflects the influence of buyer power on the retail price, and researchers have argued about the validity of this hypothesis. Dobson and Waterson and von Ungern-Sternberg showed that an increase in buyer power leads to a decline in the retail price under a situation of intense downstream competition [6,7]. Chen measured buyer power using the profit sharing capacity of leading retailers by a theoretical model, including an upstream monopoly supplier, a downstream leading retailer and several marginal retailers [4]. The results showed that buyer power may lead to a decrease in the retail price, but the impacts on the total surplus were not ascertained. Erutku extended Chen's model by introducing price competition between chain retailers and local retailers, and found that buyer power would reduce the retail price of chain retailers, but the impact on the retail price of local retailers was uncertain [4,39]. Christou and Papadopoulos also extended Chen's model, and found

that the role of buyer power is neutral, neither playing a role in raising the price nor lowering it [4,40]. Matsushima and Yoshida introduced the promotion of the leading retailer, further expanding Chen's model, and claimed that an increase in buyer power would reduce the retail price [4,41]. Chen et al. found that buyer power would reduce retail price and improve social welfare. They also examined the impact of competition, in the retail market, on buyer power, and found that the weaker the competition in the retail market, the stronger the effect of buyer power [15]. Most researchers believe that the Galbraith hypothesis is indeed true, and that buyer power may reduce the retail price under certain conditions. However, Caprice and Shekhar believe that the buyer power of retailers could increase the retail price because of the shopping cost for consumers [42]. Gaudin confirmed that the merger of downstream retailers tended to raise the retail price, but whether buyer power could reduce the wholesale price of intermediate products depended on the pass-through rate of the retail price [5]. Wang argued that buyer power would increase the retail price within a bilateral oligopoly market with two-part tariffs [43]. These studies also show that buyer power influences economic and social sustainability by influencing consumer welfare and social welfare.

In addition, many researchers have studied the effects of buyer power on manufacturers' innovation. Inderst and Wey analyzed the influence of buyer power on supplier innovation incentives, measuring buyer power using the external choice value of retailers, and found that an increase in buyer power promoted upstream innovation [16] because upstream manufacturers could improve innovation decisions by reducing the retailers' external choice value. Inderst and Shaffer and Faulí et al. measured buyer power in terms of downstream market concentration and found that buyer power could promote the innovation of upstream companies [17,18]. Chen concluded that an increase in buyer power could decrease the diversity of manufacturers' products [19] (pp. 17–40). Battigalli et al. found that buyer power reduced the quality of innovation incentives of manufacturers and damaged social welfare [20]. If buyer power has a negative impact on the process innovation and product innovation of upstream suppliers, then the buyer power of retailers will also have a negative influence on environmental and social sustainability. For example, the way the market power of large supermarkets can affect the health of consumers [9], and the way the market power of large enterprises can help improve environmental sustainability in society [10,11].

That said, these previous studies on buyer power did not consider the influence of a competition environment in the downstream market. Enterprises with buyer power are always in a complex market environment, which impacts the effects of buyer power. In addition, the literature mentioned above is mostly based on foreign retail markets and there is a lack of research on China's domestic market. As a transforming market economy, many industries in China have their own inherent characteristics. These localized market characteristics determine that the localized effects of buyer power, which may be different from those in foreign markets. This paper analyzes, based on background knowledge of China's retail market, the effects of buyer power on the wholesale price and retail price when retailers compete in the downstream market.

## 3. Market Characteristics and Basic Model

The purpose of this paper is to examine the effects of buyer power on wholesale and retail prices, and the interaction between competition and the price effects of buyer power under the conditions of downstream competition. Buyer power in the retail industry is common in the real-life economy, and the existing literature on buyer power has mostly been based on the retail industry [4,15]. This paper analyzes the influence of buyer power on price decisions, based on the background of China's retail market.

The retail market in China was dominated by a single form of department store before joining the WTO, and the scale of operation was small. When China was admitted to the WTO, foreign capital entered China's retail market, especially by means of large foreign retailers whose entry has produced demonstration and spillover effects on local retail enterprises [44,45]. This has brought about profound changes in the local retail market. These changes can be summarized into two aspects:

changes in retail formats and the increase in market concentration. The entry of foreign capital has transformed the retail market from a single form of department store to a coexistence of various forms of stores, such as department stores, supermarkets, specialty stores, and shopping malls. In addition, the admission of large retailers has created competitive pressures for local retailers, impelling mergers and restructuring, which promoted the concentration of the retail market into, for example, the China Resources Vanguard, Gome, Suning, and so forth. These large-scale retailers have strong buyer power compared with upstream manufacturers.

In addition, the buyer power of the pharmaceutical industry and the coal-fired power industry in the Chinese market is also very prominent. In the pharmaceutical industry, as a special commodity, the choice of patients for drugs is mainly determined by a professional doctor, which makes the doctor the agent of the patient and become the decision maker of whether the drug can be used by the patient. This role is similar to the retailers' "gatekeeper", and the role is prominent in the pharmaceutical industry. This position of the doctor gives them buyer power relative to the upstream pharmaceutical companies. The buyer power possessed by doctors and medical institutions has had an important impact on drug prices. In the coal-fired power industry, the contradiction between coal and electricity in the Chinese market has always been an issue plaguing enterprises and government, and it has restricted sustainable economic development. In China, power companies account for 50% of the total demand for coal, and these demands are mainly concentrated in the five major power generation groups (China Huaneng Group Corporation, China Datang Corporation, China Huadian Corporation, China Guodian Corporation, China Power Investment Corporation), the coal–electricity market is a typical monopsony market structure. The buyer power of power generation companies is also an important factor causing coal–electricity conflicts.

In order to describe the competitive structure of China's retail market, we assumed that there were two retailers in the retail market, shown as $R_1$ and $R_2$. The market structure of the two competing retailers reflected competition to simplify the model calculations, which can be extended easily to incorporate multiple retailers. Suppose there is a manufacturer $M$ in the upstream market who produces a final product, which is then resold to the final consumer by retailers $R_1$ and $R_2$. The three-tier vertical market structure consists of the manufacturer, retailer, and consumer, as shown in Figure 1.

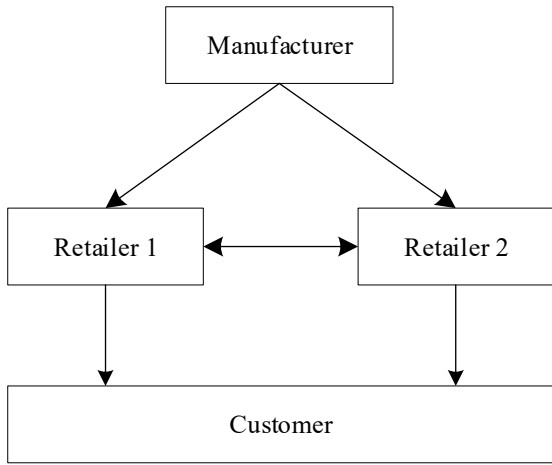

**Figure 1.** The vertical market structure shown with downstream competition.

The existing literature mainly describes the competition in the retail market through three models (as shown in Table 1). The first method, considering dominant and marginal retailers, appeared in the earlier research of Chen [4], who believed that the development of the American retail market was polarized. In the second method, competition with *N* retailers, Dobson and Waterson and von Ungern Sternberg described the competitive mode of *N* differentiated retailers with similar scales [6,7]. Changes in the number of retailers can reflect the retailer's buyer power. The third method, the duopoly

retailer competition model, similar to that of this paper, was also used. We believe that the first two models are not suitable for the structure of China's retail market. Although the concentration of China's retail market has increased over the past few decades, the market concentration is still low compared with developed countries. According to the statistics of China's Chain Store and Franchise Association (CCFA), the sales volume of China's top 100 chain enterprises reached 2.2 trillion in 2017, accounting for 6.0% of the total retail sales of consumer goods [46], while in 2010 this proportion was 11% [47], indicating that the market share of China's large retailers had not increased. There was no serious polarization phenomenon and there was no dominant individual or group of large retailers in the market. However, the market share of the four largest retailers in the UK reached 65% in 2002 [48] (p. 108), and the top five retailers in developed countries such as Austria, Belgium, Finland, Portugal and Sweden accounted for more than 60% [49] (p. 113). The second model was also unable to reflect the difference in scale of the foreign retailers, large local retailers, and other retailers. China's retail market competition shows two distinct characteristics: regional competition coexisting with national competition, and hierarchical competition. Larger local retailers compete with national retailers in different provinces and even cities. Large retailers coexist with small community supermarkets and grocery stores, forming a two-tiered competitive structure divided by scale-orientation and service.

**Table 1.** Summary of modeling approaches of retail competition in the existing literature.

| Market Competition Structure | Literature |
| --- | --- |
| Dominant and marginal retailers | Chen, Christou and Papadopoulos, Matsushima and Yoshida, Caprice and Shekhar [4,40–42] |
| Competition with $N$ retailers | Gaudin, Dobson and Waterson, von Ungern Sternberg, Chen et al. [5–7,15] |
| Duopoly retailer competition | Erutku, Inderst and Shaffer, Battigalli et al., Han et al., Gabrielsen and Johansen [17,20,39,50,51] |

Supposing a manufacturer, $M$, produces input at a constant marginal cost, $c$, the manufacturer then sells to retailers $R_1$ and $R_2$ at the liner wholesale price $w_1$ and $w_2$ respectively. The retailer's selling cost per unit product is $c_R$, standardizing $c$ and $c_R$ to zero in order to simplify the analysis. This standardized approach does not affect the core conclusions and is common in existing research. There are a large number of consumers in the market who buy products from retailers $R_1$ and $R_2$. These consumers believe that $R_1$ and $R_2$ are differentiated retailers, and the difference is not due to the physical properties of the products, but to the retailer's services, location and consumer preferences, among other factors. The utility of a consumer in purchasing products can be described by a simplified square utility function [52] (p. 36):

$$u(q_1, q_2) = q_1 + q_2 - \tfrac{1}{2}(q_1^2 + q_2^2 + 2\delta q_1 q_2) + X$$
$$s.t. \quad p_1 q_1 + p_2 q_2 + X \le I, \tag{1}$$

where $p_i$ and $q_i$ represent, respectively, the price and quantity of products purchased, by the consumer, from retailer $R_i$ ($i, j$ = 1, 2). $X$ is the utility of the consumer in terms of consuming other products and $I$ represents the consumer's income. Parameter $\delta \in (0, 1)$ represents the intensity of downstream competition; the greater the value of $\delta$, the more intense the competition in the retail market, and vice versa.

From Equation (1), we have the inverse demand function of $R_i$:

$$p_i = 1 - q_i - \delta q_j, i \ne j. \tag{2}$$

From Equation (2), we have the demand function of $R_i$:

$$q_i(p_i, p_j) = \frac{1 - \delta - p_i + \delta p_j}{1 - \delta^2}. \tag{3}$$

We assumed that neither $R_1$ nor $R_2$ has market power. This meant that the retailer's buyer power was weak enough to have an impact on the market behavior of other enterprises, because the buyer and seller power always come in pairs, and there is no possibility of an enterprise with no buyer power.

There is a two-stage game process between the manufacturer and the retailers:

- In the first stage, manufacturer, *M*, sets the wholesale price $w_1$ and $w_2$ for $R_1$ and $R_2$ according to their own profit maximization, and retailers purchase products at this wholesale price.
- In the second stage, $R_1$ and $R_2$ simultaneously set retail prices $p_1$ and $p_2$, respectively, according to their own profit maximization.

It is worth noting that the decisions of $R_1$ and $R_2$ are symmetrical. $R_1$ and $R_2$ set their retail price after observing the wholesale price of the competitor. The manufacturer and retailers make decisions based on their own profit maximization, regardless of the impact on the other side. This is the connotation of the non-cooperative model of upstream and downstream enterprises, which leads to double marginalization. We assumed that retailers could browse competitor's contracts, but retailers' contracts are often, in reality, private. In this case, manufacturers generate opportunistic behavior, which makes corporate decisions more difficult, affecting the research on the effects of buyer power. In order to rule out this interference, it is often assumed in the literature that the contract is public, for example in the papers by Horn and Wolinsky, Iozzi and Valletti, and Gaudin [5,53,54].

The game is then solved by backward induction. In the last stage of the game, the retailer takes the wholesale price as given, and sets the retail price. The decision of $R_i$ can be expressed as:

$$\max_{p_i} \pi_{R_i} = (p_i - w_i) q_i(p_i, p_j). \tag{4}$$

Bringing Equation (3) into Equation (4), and solving for the first-order condition of the retailer's profit maximization problem, we have:

$$p_i(w_i, w_j) = \frac{(2 + \delta)(1 - \delta) + 2w_i + \delta w_j}{4 - \delta^2}. \tag{5}$$

From Equation (5), we see that the retail price decision is symmetrical in the absence of buyer power, and the retail price increases with the individual wholesale price and that of the competitor. The wholesale price will naturally drive the increase of the retail price because it is a retailer's cost. The retail price also increases with the competitor's wholesale price due to the complementarity of price decisions [55].

In the first-stage of the game, the manufacturer's decision is determined by solving the following maximization problem:

$$\max_{w_i} \pi_M = \sum_{i=1}^{2} w_i q_i(w_i, w_j), \tag{6}$$
$$s.t. \quad q_i \in \operatorname{argmax} \pi_{R_i}.$$

By solving Equation (6), we found that the wholesale price of the market was $w_1 = w_2 = 1/2$. We noted that retailers pay the same wholesale price in the absence of market power. Bringing $w_1 = w_2 = 1/2$ to Equation (5), we obtained the retail price $p_1 = p_2 = (3 - 2\delta)/2(2 - \delta)$. It was clear, via the retail price, that $R_1$ and $R_2$ had the same retail price, and that retail price declined as the market competition became more intense.

## 4. Price Effects of Buyer Power

### 4.1. The Influence of Buyer Power on the Wholesale Price

In order to describe the situation in which the retailer has buyer power, we assumed that retailer $R_1$ had buyer power and $R_2$ had no buyer power. This may happen in real life if $R_1$ is a national chain retailer and $R_2$ is a local retailer. In one market, the demand for the national retailer and local retailer was symmetrical. However, due to the large scale of procurement, national retailers often adopt nationally unified procurement models, so that buyer power comes into being. Meanwhile, local retailers purchase products individually and the procurement scale is limited, therefore buyer power is not formed. No matter which market is selected, the competition mode between retailers can be described by Figure 1. For ease of distinction and narration, $R_1$ is hereafter referred to as a national retailer, and $R_2$ as a local retailer.

In the case that $R_1$ has buyer power, the decision-making process between the manufacturer and retailers is a three-stage game:

- In the first stage, manufacturer $M$ and national retailer $R_1$ negotiate to determine the wholesale price $w_1$, and the wholesale price $w_2$ is set by the manufacturer for local retailer $R_2$.
- In the second stage, national retailer $R_1$ sets retail price $p_1$.
- In the third stage, local retailer $R_2$ sets retail price $p_2$.

Amir and Stepanova [56] analyzed the issue of endogenous timing and first- versus second-mover advantage in differentiated-product Bertrand duopoly with asymmetric costs. This yielded a unique outcome of sequential play with the more efficient firm as leader and the less efficient firm as follower. The second-mover advantage, of the follower, had no effect on the endogenous timing. Therefore, it is reasonable for us to set up the national retailer as a price leader and the local retailer as a follower. The essential difference between this paper and Erutku is the three-stage game. Erutku used a two-stage game to allow $R_1$ and $R_2$ to set the retail price simultaneously [39].

Using backward induction to solve the enterprises' decision-making problem, assuming that the national retailer has buyer power, in the last stage of the game $R_2$ takes the wholesale price, $w_2$, as given, and sets the retail price $p_2$. Its profit is thus given by:

$$\max_{p_2}\pi_{R_2} = (p_2 - w_2)q_2(p_2, p_1). \tag{7}$$

From Equation (7), we have:

$$p_2(p_1, w_2) = \frac{1 - \delta + \delta p_1 + w_2}{2}. \tag{8}$$

Comparing Equations (8) and (5), it can be seen that when $R_1$ has buyer power, $R_2$'s retail price depends not only on its wholesale price, but also on the retail price of $R_1$. The pricing decision of the local retailer is decided by the national retailer's price. In general, national retailers are large and dominate the market, and other retailers take actions based on the decisions of these retailers, such as Wal-Mart, Tesco, and Suning.

In the second stage of the game, the decision of $R_1$ is:

$$\max_{p_1}\pi_{R_1} = (p_1 - w_1)q_1(p_1, w_2). \tag{9}$$

Solving for market equilibrium, the retail price is:

$$p_1(w_1, w_2) = \frac{(2 + \delta)(1 - \delta) + (2 - \delta^2)w_1 + \delta w_2}{2(2 - \delta^2)}, \tag{10}$$

$$p_2(w_2, w_1) = \frac{(1-\delta)(4+2\delta-\delta^2)+(4-\delta^2)w_2+\delta(2-\delta^2)w_1}{4(2-\delta^2)}. \tag{11}$$

The wholesale price decision takes place in the first stage. The manufacturer $M$ sets the wholesale price $w_2$ for local retailer $R_2$, and $R_2$ can only accept it passively. At the same time, the manufacturer $M$ and the national retailer $R_1$ negotiate a wholesale price $w_1$. The wholesale price of $R_2$ can be used as a benchmark for $R_1$, who only needs to negotiate a discount. Assuming $w_1 = (1-\gamma)w_2$, $\gamma \in (0,1)$ represents the discount received by $R_1$, which was used for measuring buyer power. The bigger $\gamma$, the greater the discount and the stronger the market power. We assumed that both retailers remained in the market. Therefore, the discount conditions proposed by $R_1$ did not break through the bottom line of $M$, and, for the manufacturer, it was more profitable to trade with two retailers. The first-stage equilibrium was determined by solving the following maximization problem:

$$\max_{w_1, w_2} \pi_M = w_1 q_1(p_1(w_1, w_2), w_2) + w_2 q_2(p_1(w_1, w_2), w_2),$$
$$s.t. \quad w_1 = (1-\gamma)w_2, \tag{12}$$
$$\pi_M \geq 1/8.$$

where $\pi_M = 1/8$ is the profit when $M$ trades only with $R_2$. The equilibrium wholesale price is solved when $R_1$ has buyer power:

$$w_1^* = \frac{(1-\gamma)(1-\delta)[8+4\delta-3\delta^2-\delta^3-(4+2\delta-2\delta^2-\delta^3)\gamma]}{2[(2-\delta^2)\gamma^2-2(1+\delta)(4-4\delta+\delta^3)\gamma+8-4\delta-7\delta^2+2\delta^3+\delta^4]}, \tag{13}$$

$$w_2^* = \frac{(1-\delta)[8+4\delta-3\delta^2-\delta^3-(4+2\delta-2\delta^2-\delta^3)\gamma]}{2[(2-\delta^2)\gamma^2-2(1+\delta)(4-4\delta+\delta^3)\gamma+8-4\delta-7\delta^2+2\delta^3+\delta^4]}. \tag{14}$$

**Proposition 1.** *As there is an increase in the buyer power of the national retailer, its own wholesale price declines, and the local retailer's wholesale price first rises and then falls.*

**Proof.** The proof of Proposition 1 is given in Appendix A. □

In non-cooperative trading mode, a national retailer will ask the manufacturer for a wholesale price discount according to their own market power. As there is an increase in buyer power, the wholesale price of the national retailer, understandably, decreases. However, why is the effect of buyer power on the local retailer uncertain?

Considering the wholesale price of $R_2$ as a benchmark, the process by which the manufacturer sets the benchmark wholesale price $w_2$ is, on the one hand, to obtain profits from retailers and, on the other hand, to respond to the influence of buyer power. In order to analyze the manufacturer's mechanism to respond to buyer power, we assumed the benchmark wholesale price was constant and analyzed the influence of buyer power on the manufacturer's profit, as well as the manufacturer's response to these effects. Based on Equation (12), by taking the partial derivative of $\pi_M$ with respect to $\gamma$, we established the following:

$$\frac{\partial \pi_M}{\partial \gamma} = \underbrace{\frac{\partial(-\gamma w_2 q_1)}{\partial \gamma}}_{\text{Profit Reduction}} + \underbrace{w_2 \left(\frac{\partial q_1}{\partial \gamma} - \left|\frac{\partial q_2}{\partial \gamma}\right|\right)}_{\text{Demand Transfer}}. \tag{15}$$

Equation (15) indicates that buyer power will affect the manufacturer's profit in two ways:

1.  Profit reduction: The increase in buyer power reduces the wholesale price paid by the national retailer $\gamma w_2$, and reduces the manufacturer's profit obtained from the national retailer $\gamma w_2 q_1$. We call this the profit reduction effect.

2. Demand transfer: The difference in wholesale price between the national and local retailer is $\gamma w_2$. The local retailer's higher wholesale price leads to a higher retail price, causing some consumers to shift from the local retailer to the national retailer. Remember that the manufacturer earns lower margins from the national retailer, resulting in the transfer of market demand, which also reduces the profit of the manufacturer—this is called the demand transfer effect.

The manufacturer's response to these two effects is exactly the decision-making process of the benchmark wholesale price. The manufacturer can respond to the profit reduction effect by raising the benchmark wholesale price. Similarly, the demand transfer effect can be dealt with by reducing $w_2$ to decrease the cost disadvantage of the local retailer. The ultimate size of the benchmark wholesale price depends on the strength of these two effects. When buyer power is weak, the retailers' cost difference $\gamma w_2$ is small. Therefore the demand transfer effect becomes weak and the manufacturer can only reduce the profit reduction effect by increasing the benchmark wholesale price. However, when buyer power increases to a certain extent and the retailers' cost difference is large, the 'demand transfer effect' is relatively strong. If the manufacturer raises the wholesale price, the 'demand transfer effect' is further enhanced. In this case, the manufacturer's decision-making objective should be changed from alleviating the 'profit reduction effect' to reducing the 'demand transfer effect'. In other words, the manufacturer should reduce the benchmark wholesale price. Therefore, the wholesale price of the local retailer will first rise and then fall.

Comparing the game order with Section 3, we found that the buyer power of $R_1$ was mainly reflected in two aspects:

1. The sequence of retailers' decision-making was different. When $R_1$ had buyer power, $R_1$ dominated retail price decisions, and the price decision of $R_1$ and $R_2$ was similar to Stackelberg. In the absence of buyer power, the two retailers set the optimal retail price simultaneously.
2. Wholesale prices were determined in different ways. When $R_1$ had buyer power, the manufacturer and $R_1$ negotiated the wholesale price. In the absence of buyer power, the manufacturer set the wholesale price for $R_1$ and $R_2$ according to their own profit maximization.

The former is the embodiment of buyer power in horizontal competition, referring to the influence of buyer power on the retailer's price decisions. We call it the 'horizontal effect'. The latter reflects the buyer power in the relationship between upstream and downstream, representing the influence of buyer power on the manufacturer's price decisions, which is called the 'vertical effect'. The mechanism by which buyer power affects the wholesale price also includes the 'horizontal effect' and the 'vertical effect'. This is another difference between this paper and Erutku. We analyzed the direct vertical effect and indirect horizontal effect of buyer power, while Erutku only considered the vertical effect [39].

In order to eliminate the influence of the horizontal effect on the wholesale price decision, we considered an 'assumption scenario' in which the manufacturer and the retailer held a two-stage game. This method was the same as Erutku [39]:

- In the first stage, the manufacturer set the wholesale price $w_2$ for $R_2$ and the manufacturer negotiated a wholesale price $w_1$ with $R_1$.
- In the second stage, $R_1$ and $R_2$ set the retail prices $p_1$ and $p_2$ simultaneously.

In the game of the 'assumption scenario', the first stage is the same as that in which the national retailer has buyer power, and the second stage is the same as when there is an absence of buyer power. Table 2 gives a comparison of the game processes in three different situations.

There is no horizontal effect under the assumption scenario, so the direct influence of buyer power on the wholesale price will appear. We call it the vertical effect of buyer power. Solving the game under the assumption scenario, the manufacturer's optimal wholesale price is:

$$w_2^{s*} = \frac{(2+\delta)(1-\delta)(2-\gamma)}{(2-\delta^2)\gamma^2 + 2(2+\delta)(1-\delta)\gamma - 2(2+\delta)(1-\delta)}. \tag{16}$$

**Table 2.** Comparison of game processes in different situations.

| Game Process | | No Buyer Power | Buyer Power | Assumption Scenario |
|---|---|---|---|---|
| Wholesale price | First stage | $M$ sets $w_1,w_2$ for $R_1$ and $R_2$ | $M$ sets $w_2$ for $R_2$, negotiates $w_2$ with $R_1$ | $M$ sets $w_2$ for $R_2$, negotiates $w_2$ with $R_1$ |
| Retail price | Second stage | $R_1$ and $R_2$ set $p_1$ and $p_2$ simultaneously | $R_1$ sets $p_1$ | $R_1$ and $R_2$ set $p_1$ and $p_2$ simultaneously |
| | Third stage | NA | $R_2$ sets $p_2$ | NA |

From Equation (16), we made the following inferences.

**Corollary 1.** *In a downstream competition market, if the horizontal effect of buyer power is not taken into account, as there are increases in the buyer power of the national retailer, the wholesale price of the national retailer declines, and the wholesale price paid by the local retailer rises first and then falls. This is the same as the conclusion of Proposition 1.*

Based on the above analysis, the wholesale price is affected by buyer power in two channels (see Figure 2). In Channel 1, buyer power affects the manufacturer's profit, which in turn leads to the change of the manufacturer's wholesale price, which is called the vertical effect of buyer power. Buyer power changes the retailers' decision model in Channel 2, affecting the manufacturer's wholesale price decision and leading to a change in the wholesale price, which is called the horizontal effect of buyer power.

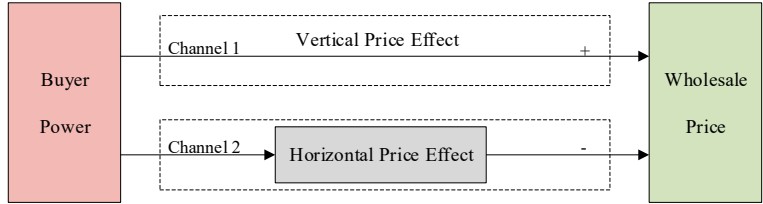

**Figure 2.** The channels of buyer power influencing wholesale price.

In the real economy, the two channels in Figure 2 may come into being in different market scenarios. Channel 2 may occur in situations where buyer power is greater, while Channel 1 may occur in the complete opposite conditions and not affect the retailer's horizontal decision model. In addition, a concern about buyer power is that an increase in buyer power may raise the trading price of competitors, thereby changing the competitive structure of the retail market and crowding out competitors. The conclusion of Proposition 1 shows that buyer power raises the wholesale price of the local retailer only if buyer power is small. Once buyer power is strong enough, the manufacturer lowers the wholesale price of the local retailer to enhance its competitiveness. Although increased buyer power does affect the competitive structure of the retail market, which may be detrimental to the local retailer, this situation does not continue and the probability of the local retailer being squeezed out of the market is small.

### 4.2. The Influence of Buyer Power on Retail Price

The influence of buyer power on the retail price includes the horizontal and vertical effect. First, buyer power transforms the competition of enterprises into a leader-follower model, which reduces the degree of competition, and may lead to an increase in the retail price. When national retailers have no buyer power, the competition among retailers in the market is similar to Bertrand. If the national retailers are dominant and have market power, the retailers' price decisions are similar to Stackelberg.

Bertrand is more competitive than Stackelberg, so retailers have a higher retail price in the case of having buyer power. Second, buyer power indirectly affects the retail price by impacting the wholesale price. On the whole, the influence of buyer power on the retail price is the result of both the horizontal and vertical effect.

Bringing Equations (13) and (14) into Equations (10) and (11), we obtained the final retail price:

$$p_1^* = \frac{(2+\delta)(1-\delta) + (2-\delta^2)w_1^* + \delta w_2^*}{2(2-\delta^2)}, \tag{17}$$

$$p_2^* = \frac{(1-\delta)(4+2\delta-\delta^2) + (4-\delta^2)w_2^* + \delta(2-\delta^2)w_1^*}{4(2-\delta^2)}. \tag{18}$$

**Proposition 2.** *In the case of downstream competition:*

1.  *When buyer power is small, the emergence of buyer power will increase the retail price compared with a scenario in which there is an absence of buyer power. When buyer power is large, the retail price decreases.*
2.  *With an increase in buyer power, the retail price of the national retailer declines while the retail price of the local retailer rises first and then falls, and the average price decreases.*

**Proof.** The proof of Proposition 2 is given in Appendix B. □

Proposition 2 consists of two conclusions. The first uses a comparison of the retail prices, one in which the retailer has buyer power versus one in which the retailer has no buyer power. The second conclusion describes the change of retail price when the retailer has buyer power, as buyer power becomes stronger from a state of weakness. There are some differences between these two conclusions. Firstly, the nature of the comparison is different. The former compares the situation between having buyer power and having no buyer power, which is a qualitative change, while the latter analyzes the change of buyer power, which is a quantitative change. Secondly, the core factors causing the differences are different. In the case in which there is buyer power, both horizontal and vertical effects have an impact on the retail price, while the retail price is mainly affected by the vertical effect in the latter analysis.

In order to understand Proposition 2 more deeply, we first analyzed Part 2 of Proposition 2. It can be seen from Equations (10) and (11), $\partial p_i / \partial w_i > \partial p_i / \partial w_j > 0$ that changes in the individual wholesale price of itself and that of the competitor have an impact on the retail price, and the impact of the individual wholesale price is stronger. According to Proposition 1, buyer power causes the wholesale price of the national retailer to fall, and the wholesale price of the local retailer to rise first and then fall, both of which have an impact on the retail price. For national retailers' retail price, the decline in their own wholesale price leads to a fall, and changes in competitors' wholesale prices lead to a rise first and then a fall. Since the effect of its own is stronger than that of the competitor, with the increase in buyer power, the retail price of national retailers tends to decline. Similarly, the retail prices of local retailers rise first and then fall as buyer power increases. Generally speaking, higher sales of the national retailer lower the overall retail price, and the average retail price is apt to decline.

Next, we analyzed Part 1 of Proposition 2 to discuss the changing mechanism of the retail price in the case of a national retailer with and without buyer power. In the shift from no buyer power to buyer power, there were three channels affecting the retail price. Firstly, the horizontal effect of buyer power was the direct way for buyer power to influence retail price. Since Bertrand was more competitive than Stackelberg, retailers had a higher retail price in the case of buyer power, and this channel tended to increase retail price. The second channel was the indirect effect of the vertical effect of buyer power. Namely, this channel had an indirect impact on the retail price by affecting the wholesale price, which tended to lower the retail price. Thirdly, the indirect effect of the horizontal effect of buyer power was the same as the second channel. In other words, it affected the retail price by

affecting the wholesale price. When buyer power was weak, the first channel was dominant and the retail price rose. When buyer power was strong, the impact created by the latter two channels became more profound, and held a dominant position over Channel 1, leading to a decrease in the retail price. These three channels in which buyer power influences the retail price are represented by Figure 3.

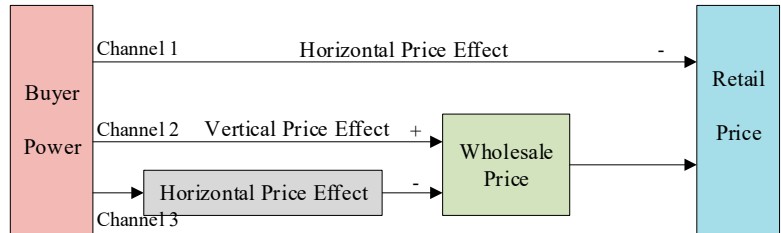

**Figure 3.** The channels of buyer power influencing the retail price.

## 5. The Impact of Downstream Competition on the Price Effects of Buyer Power

We examined the influence of buyer power on the wholesale price and retail price in the case of downstream competition, as described above. Then, we examined the impact of downstream competition on the price effect of buyer power. This is another major difference between this paper and Erutku in that he did not consider the impact of retailer competition on the price effect of buyer power [39].

First we analyzed the impact of downstream competition on buyer power's wholesale price effect. In the absence of market power, the wholesale price paid by the national retailer and local retailer was $w_1 = w_2 = 1/2$, and the wholesale price had nothing to do with competition. In the case in which $R_1$ had market power, the wholesale price paid by retailers depended on the degree of competition. In order to illustrate the impact of competition on the wholesale price more clearly, we simulated the trend of the wholesale price with intensive competition, when the national retailer's buyer power was $\gamma \in \{0.2, 0.4, 0.6, 0.8\}$. This is shown in Figure 4a–d. The situations $\gamma = 0.6$ and $\gamma = 0.8$ are unlikely to happen in real life because we measured the national retailer's buyer power using its wholesale price discount. The purpose of doing this was simply to theoretically explain the impact of competition on the buyer power's price effect.

From Figure 4, we see that the wholesale price of the national retailer (blue line) is always less than 0.5; that is, the buyer power reduces the wholesale price of the national retailer, which is consistent with Proposition 1. With the increase in competition, the wholesale price of the national retailer decreases. This shows that fierce competition in the retail market makes buyer power further reduce the national retailer's wholesale price.

In Figure 4a, when $\gamma = 0.2$, the wholesale price of the local retailer (red line) is always bigger than 0.5, no matter the level of competition. In Figure 4b, when $\gamma = 0.4$, the wholesale price paid by the local retailer is less than 0.5 as the competition increases. A similar situation can be seen in Figures 4c and 4d. For the above analysis, we know that buyer power influences the wholesale price decisions of the manufacturer through the 'profit reduction effect' and 'demand transfer effect'. Market competition enhances the 'demand transfer effect' but weakens the 'profit reduction effect' of buyer power. As market competition becomes fierce, the wholesale price gradually decreases. This indicates that the 'demand transfer effect' occupies a dominant position, which can be seen in Figure 4.

Next, we analyzed the impacts of competition on the retail price effect of buyer power. Using the same approach, we simulated the relationship between competition and the retail price of the national retailer and local retailer when $\gamma = 0.2$ and $\gamma = 0.4$, which is shown in Figure 5a–b. Since market power is expressed by the proportion of the wholesale price discount obtained by the national retailer, we did not simulate a situation with larger market power in order to be more realistic.

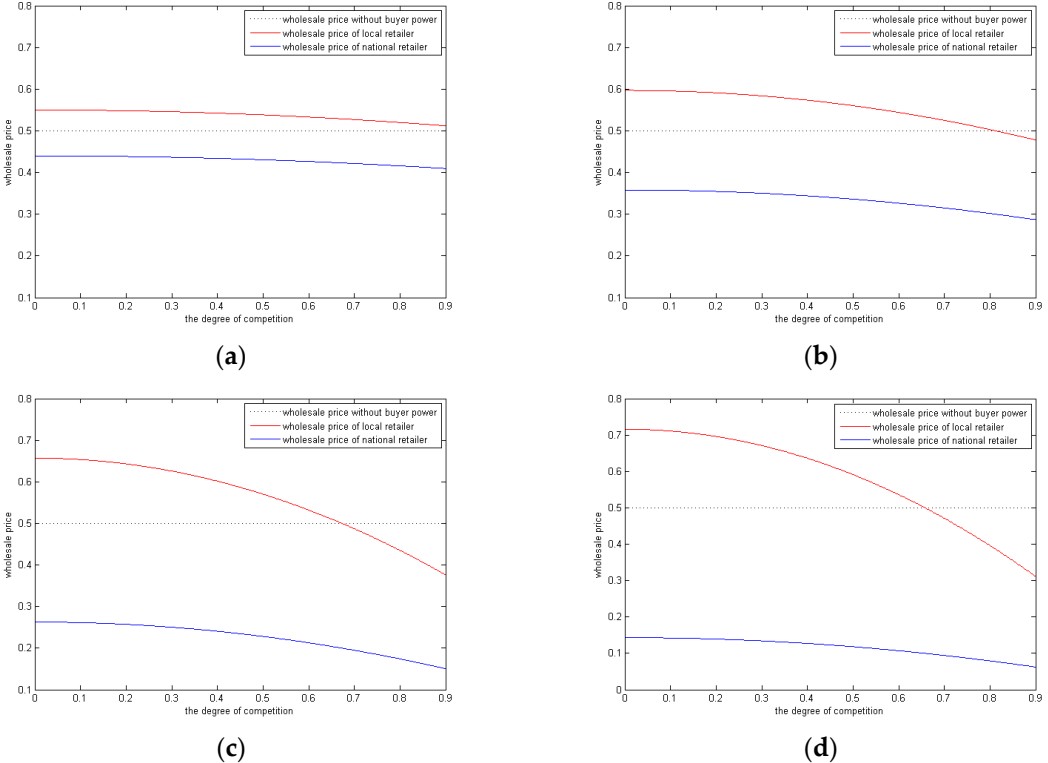

**Figure 4.** (**a**) The relationship between the wholesale price and the degree of competition when $\gamma = 0.2$; (**b**) the relationship between the wholesale price and the degree of competition when $\gamma = 0.4$; (**c**) the relationship between the wholesale price and the degree of competition when $\gamma = 0.6$; and (**d**) the relationship between the wholesale price and the degree of competition when $\gamma = 0.8$.

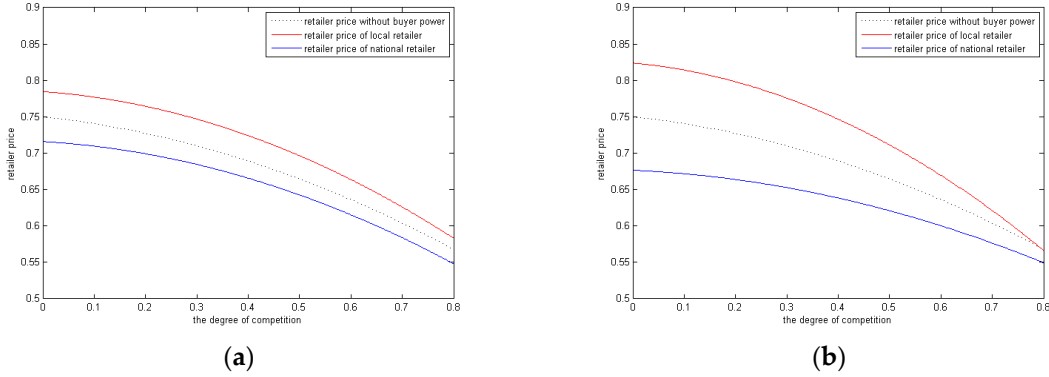

**Figure 5.** (**a**) The relationship between competition and retail price when $\gamma = 0.2$ and (**b**) the relationship between competition and the retail price when $\gamma = 0.4$.

As we can see from Figure 5, the effect of market competition on the retail price is similar to that of the wholesale price:

1.  The retail price declined as competition increases;
2.  Compared with no buyer power, the retail price of the national retailer was lower in the case of buyer power, while the retail price of the local retailer was greater;
3.  With the increase of retail market competition, the gap between the retail price with buyer power and without buyer power shrunk, indicating that the difference caused by buyer power was reduced, and that market competition and buyer power had an alternative effect on retail price.

In addition, previous literature has indicated that an increase in buyer power reduces the retail price in the case of fierce competition among downstream retailers [6,7]. Comparing the two images in Figure 5 we found that, when buyer power was small (Figure 5a), as competition increased, the price of the local retailer was always above the dotted line, representing the retail price without buyer power, and the national retailer's price is below this. However, when buyer power was strong (Figure 5b), the retail price of the local retailer appeared below the dotted line. In other words, in the case of buyer power being strong, with an increase in competition, the retail price of the local retailer became lower than that without buyer power. This conclusion is consistent with Proposition 2, and complements the existing conclusions of [6,7].

## 6. Conclusions

Based on the description of the competitive structure of China's retail market, this paper built a market structure model of competition between national retailers and local retailers, examining a decision problem under the assumption that national retailers either have buyer power or no buyer power, analyzing the impact of buyer power on the wholesale price and retail price, and revealing the underlying mechanisms behind the influence of buyer power. This paper also examined the impact of downstream competition on the price effect of buyer power, and the interaction between competition and buyer power. The findings of the research are as follows:

(1) The retailer's wholesale price was affected by buyer power in two ways. Firstly, the increase in buyer power of the national retailer changed the structure of the downstream market, having an impact on the profit of the manufacturer. The manufacturer could respond by adjusting the wholesale price. This channel expressed a direct influence of buyer power on the wholesale price, which is called the vertical effect of buyer power in this paper. Secondly, buyer power could influence the profit of the manufacturer by changing market demand, and the manufacturer could respond by adjusting the wholesale price. We called this the horizon effect of buyer power, which is an indirect effect.

(2) The mechanism by which buyer power influences the retail price can also be divided into two aspects. Firstly, buyer power reduces the market competition, and this horizontal effect tends to reduce the retail price. Secondly, the influence of buyer power on the wholesale price affects the retail price owing to the fact that the retail price is a function of the wholesale price. Buyer power was able to influence the wholesale price in two different ways, coupled with the horizontal effect on the retail price; therefore, buyer power mainly affected the final retail price in three channels.

(3) An increase in buyer power reduced the national retailer's wholesale price, but the influence on the local retailer was uncertain. The wholesale price of the local retailer first rose and then fell with the increase of the national retailer's buyer power. An analogous result holds true for the retail price. On the whole, the increase in buyer power reduced the average retail price in the market and improved consumer welfare.

(4) Regarding the interaction between buyer power and downstream competition, our research found that downstream competition enhanced the horizontal effect of buyer power, affecting the wholesale price, but weakened the vertical effect. The competition of the retail market and the role of buyer power was substitutable. When the retail market was highly competitive, the influence of buyer power on the retail price shrunk. Buyer power, in turn, affected the competitive advantage of the national retailer and local retailer, but the local retailer could not be withdrawn from the market.

According to the conclusions, we were enlightened in the following ways. There are no specific provisions in China's Anti-monopoly Law to regulate buyer power. With the development of "Internet + retailers", the competition between online retailers and physical retailers, as well as large retailers, has become more and more fierce. The Internet has an impact on the traditional retail industry, but it also limits the emergence of super-large monopoly power. Therefore, in the regulation process of buyer power, we should mainly focus on the competitive structure of the downstream market and the trading characteristics of retailers. Only in this way can we promote the healthy development of the market and ensure the sustainability of development.

We list some possible extensions in this paper. Firstly, the basic model of this paper is upstream monopoly and downstream competition. In future research, we can expand to upstream competition and downstream monopoly, and competition in the upstream and downstream. Secondly, in the game model of this paper, we assumed that upstream and downstream determine the wholesale price through a non-cooperative game, which is a simplified model. We could use the bargaining model, pricing power scramble model, and mutual dependency model [57] in future research. Finally, we only analyzed market competition, which is a kind of market environment. Factors such as the number of enterprises, product differentiation, the size of enterprises, and the trading patterns between upstream and downstream enterprises also interact with market power and should be explored further in the future.

**Funding:** This research received no external funding.

**Acknowledgments:** The author is very grateful to the comments of four anonymous referees. The author also thanks the editor, Joyce Ma for her help, and Kai Li, Wei Li, and Chenxi Tang for their valuable and insightful comments and suggestions.

**Conflicts of Interest:** The author declares no conflict of interest.

## Appendix A. Proof of Proposition 1

From Equation (13), the first order partial derivatives of $w_1^*$ with respect to $\gamma$ as:

$$\frac{\partial w_1^*}{\partial \gamma} = \frac{(1-\delta)f(\delta,\gamma)}{2[(2-\delta^2)\gamma^2 - 2(1+\delta)(4-4\delta+\delta^3)\gamma + 8 - 4\delta - 7\delta^2 + 2\delta^3 + \delta^4]^2}, \tag{A1}$$

where $f(\delta,\gamma) = A\gamma^2 + B\gamma + C$, $A = \delta^6 + 6\delta^5 - 24\delta^3 - 12\delta^2 + 24\delta + 16$, $B = -(2\delta^6 + 6\delta^5 - 14\delta^4 - 28\delta^3 + 16\delta^2 + 32\delta)$, $C = \delta^6 - 15\delta^4 + 2\delta^3 + 44\delta^2 - 32$.

It's easy to determine $sign\{\frac{\partial w_1^*}{\partial \gamma}\} = sign\{f(\delta,\gamma)\}$, so we needed to judge the symbol of $f(\delta,\gamma)$. Viewing $f(\delta,\gamma)$ as a quadratic function of $\gamma$, the value at the symmetry axis and endpoint were:

$$f(\delta, -\frac{B}{2A}) = -\frac{4(1+\delta)(4-3\delta^2)(8+4\delta-3\delta^2-\delta^3)}{\delta^2 + 6\delta + 4} < 0, \tag{A2}$$

$$f(\delta, 0) = \delta^6 - 15\delta^4 + 2\delta^3 + 44\delta^2 - 32 < 0, \tag{A3}$$

$$f(\delta, 1) = -(4-3\delta^2)(4+2\delta-\delta^2) < 0. \tag{A4}$$

According to the nature of the unary quadratic function, $f(\delta,\gamma) < 0$ in $\gamma \in (0,1)$ is true, so $\frac{\partial w_1^*}{\partial \gamma} < 0$. That is, as buyer power increased the wholesale price of the national retailer fell.

Similarly,

$$\frac{\partial w_2^*}{\partial \gamma} = \frac{(1-\delta)(2-\delta^2)g(\delta,\gamma)}{2[(2-\delta^2)\gamma^2 - 2(1+\delta)(4-4\delta+\delta^3)\gamma + 8 - 4\delta - 7\delta^2 + 2\delta^3 + \delta^4]^2}, \tag{A5}$$

where $g(\delta,\gamma) = D\gamma^2 + E\gamma + H$, $D = 6\delta^5 + 2\delta^4 - 4\delta^3 - 8\delta^2 + 4\delta + 8 > 0$, $E = -(2\delta^5 + 6\delta^4 - 12\delta^3 - 28\delta^2 + 16\delta + 32)$, $H = \delta^5 + 4\delta^4 - 3\delta^3 - 18\delta^2 + 16$.

In the same way, viewing $g(\delta,\gamma)$ as a quadratic function of $\gamma$, the symmetry axis was $-\frac{E}{2D} = \frac{8+4\delta-3\delta^2-\delta^3}{(2+\delta)(2-\delta^2)}$. Through numerical simulation we had $-\frac{E}{2D} > 1$, coupled with $D > 0$, so $g(\delta,\gamma)$ was a decreased function with respect to $\gamma$ in $\gamma \in (0,1)$. We determined that

$$g(\delta, 0) = \delta^5 + 4\delta^4 - 3\delta^3 - 18\delta^2 + 16 > 0, \tag{A6}$$

$$\text{and } g(\delta, 1) = 5\delta^3 + 2\delta^2 - 12\delta - 8 < 0. \tag{A7}$$

Therefore, there is a critical value $\hat{\gamma} \in (0,1)$, when $0 < \gamma < \hat{\gamma}, g(\delta, \gamma) > 0$; when $\hat{\gamma} < \gamma < 1$, $g(\delta, \gamma) < 0$, where $\hat{\gamma} = \frac{-E + \sqrt{\Delta}}{2D}$, $\Delta = 16(1 + \delta)(8 + 4\delta - 3\delta^2 - \delta^3)(2 - \delta^2)^2$.

So we can be sure when $\gamma \in (0, \hat{\gamma})$, $\partial w_2^* / \partial \gamma > 0$; when $\gamma \in (\hat{\gamma}, 1)$, $\partial w_2^* / \partial \gamma < 0$. That is, with the increase in buyer power of the national retailer, the wholesale price of the local retailer first rose and then fell.

## Appendix B. Proof of Proposition 2

Firstly, let us prove Part 2 of Proposition 2. From Equation (17), the first order of partial derivatives of $p_1^*$ with respect to $\gamma$ was:

$$\frac{\partial p_1^*}{\partial \gamma} = \frac{(1 - \delta)(2 - \delta^2)h(\delta, \gamma)}{2[(2 - \delta^2)\gamma^2 - 2(1 + \delta)(4 - 4\delta + \delta^3)\gamma + 8 - 4\delta - 7\delta^2 + 2\delta^3 + \delta^4]^2}, \tag{A8}$$

where $h(\delta, \gamma) = J\gamma^2 + K\gamma + L, J = 4 + 8\delta - 4\delta^3 - \delta^4$, $K = 2\delta(\delta^3 + 3\delta^2 - 4\delta - 8)$, and $L = -\delta^4 - 2\delta^3 + 7\delta^2 + 4\delta - 8$. Using the same method we could prove $\frac{\partial p_1^*}{\partial \gamma} < 0$.

Similarly, it can be proved that there is a critical value $\widetilde{\gamma} \in (0,1)$, when $0 < \gamma < \widetilde{\gamma}, \frac{\partial p_2^*}{\partial \gamma} > 0$; when $\widetilde{\gamma} < \gamma < 1, \frac{\partial p_2^*}{\partial \gamma} < 0$.

If $\gamma = 0$, it is easy to prove that the retail price with buyer power is higher than the retail price without buyer power, and this is a result of the horizontal price effect. Part 1 of Proposition 2 can be proved by combining the above propositions.

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
