# Peer review of "The Influence of Buyer Power on Supply Chain Pricing with Downstream Competition"

_sustainability, doi:10.3390/su11102924_

Round 1

Reviewer 1 Report

See the attachment.

Reviewer 2 Report

the manuscript is interest d and nicely written and I think that it can be accepted in the present form

Author Response

Response to Reviewer 2 Comments

Point 1: the manuscript is interested and nicely written and I think that it can be accepted in the present form, English language and style are fine/minor spell check required

Response 1: Thank you very much for your recognition. We have adopted the English editing service provided by MDPI, and hope to meet your requirements.

Point 1: I suggest authors to be more concise in the parts prior to the model: for instance, could you provide some example from the Chinese sectors where buyer power is more evident (as in the food sector, what for example in other sectors such as robotics or textile?)

Response 1: Our research is based on the Chinese retail industry. There is no example in our manuscript, this is a pity. In the resubmitted version, we removed some of the textual narratives in front of the model to make the article more concise. At the same time, we added examples example from the Chinese sectors where buyer power is more evident.

In the retail sector, home appliance sellers Gome and Suning have strong market power for manufacturers due to their large sales network and large sales volume, and enjoy very favorable supply terms. Large-scale chain retailers also have strong bargaining power in the face of manufacturers because of unified procurement, such as Wal-Mart, Tesco, Carrefour, China Resources Vanguard, etc.

In the Chinese market, the pharmaceutical industry is also heavily influenced by the buyer power. As a special commodity, the choice of patients for drugs is mainly determined by a professional doctor, which makes the doctor become the agent of the patient. This role is similar to the retailers’ “gatekeeper” in the retail market, and the role is more prominent in the pharmaceutical industry. This position of the doctor gives it a buyer power relative to the upstream pharmaceutical companies. The buyer power possessed by doctors and medical institutions has had an important impact on drug prices.

In addition, the contradiction between coal and electricity in the Chinese market has always been an important problem that plagues enterprises and the government, and it also restricts sustainable economic development. In the Chinese market, power companies account for 50% of total demand for coal, and these demand are mainly concentrated in the five major power generation groups (China Huaneng Group Corporation, China Datang Corporation, China Huadian Corporation, China Guodian Corporation, China Power Investment Corporation), the coal-electricity market is a typical monopsony market structure. The buyer power of power generation companies is also an important factor causing coal-electricity conflicts.

Point 2: Please use at least once the economic terms of monopsony or quasi-monopsony for buyer power.

Response 2: The source and concept of buyer power have been analyzed in the article. Robinson (1933) proposed the concept of monopsony power to represent the market power of the buyer relative to the seller. This concept is proposed by the analog monopoly power, reflecting the market power of downstream monopoly buyer relative to the upstream perfect competitive seller. Galbraith proposed the concept of countervailing power in 1952. Then OECD (1981), Dobson et al. (1998), and Clarke et al. (2002) all defined the concept of buyer power. Finally, Chen (2007) summarized these concepts. The meaning of buyer power is broader. It refers to the ability of retailers to obtain lower than the normal supply price or superior supply conditions. When the upstream market is completely competitive, the buyer power of the retail market is a monopsony power, and when the upstream is incomplete competition, the buyer power is countervailing power or bargaining power. These have been revised in the article and new references have been added.

Point 3: Line 210: please provide official statistics to justify the choice of your model.

Response 3: Compared with developed countries, the market concentration of retailers is low in the Chinese retail market, although the sales scale of large retailers has increased year by year. Especially with the rise of online sales, the market share of large retailers has declined year by year. According to the statistics of China Chain Store & Franchise Association (CCFA), the sales volume of China's top 100 chain enterprises reached 2.2 trillion in 2017, accounting for 6.0% of the total retail sales of consumer goods, while in 2010 this proportion was 11%, indicating that the market share of China's large retailers has not increased, and there is no serious polarization phenomenon. However, the market share of the four largest retailers in the UK reached 65% in 2002 [UK Competition Commission, 2008], and the top five retailers in developed countries such as Austria, Belgium, Finland, Portugal and Sweden accounted for more than 60% [Dobson et al., 2003, p. 113]. These data indicate that the Chinese market is different from the previous research background and requires different models. At the same time, China has a wide variety of retailers now, including supermarkets, hypermarkets, department stores, convenience stores and specialty stores, etc., which vary greatly in the number of employees and the size of business area. It is also difficult to reflect the reality of China by using models that N retailers compete. These have also been revised in the article.

Reviewer 3 Report

The rticle is interesting and addresses a topic of great interest, relevant in the socio-economic scientific debate. 

In particular, another element of interest is represented by the fact that the study is conducted on the Chinese market, a context that represents specific socio-economic and political conditions for this type of analysis. However, the article is predominantly econometric and does not leave much spaces for argumentative considerations or for the inclusion of variables that can enhance the relevance and peculiarity of the chosen context of analysis.

The argumentative structure is effective and the research is rigorous and appreciable from a methodological point of view. 

The article is therefore publishable but we recommend some minor changes to make it suitable for the themes, purposes and approach of the journal:

-The relation between buyer power and sustainability is always recalled but non even explicitly explained in the introductory part. Is it only refers to an economic sustainbility, or the author want to refer to enviromental or social sustainability also? Reading the article it is evident that we refer mainly on the first idea of sustainability, but in the introductory part maybe the other two could be taken into account, specifing only at the end that the author will be focus his analysis on economic/market sustainability. It is clear that buyer power could be unsustainble for the suppliers (reducing margins or putting some of them outiside of the market) or customers (downloading on them the cost of the retailer buyer power) but it also could push suppliers to reduce the quality of their productive process or their product impacting negatively on the environment or the health of consumers. Despite the paper face the effect on price and competition, some consideration about the different form of impact on sustainability generated by the buyer power of mass retailers has to be mentioned and explained, mainly in the introduction or the literature review paragraph.

-It  less explained the socio-economical peculiartity of the chinese retail markets. How can we explain its charecterstics and peculiarities? What specific historical, normative or istitutional environment could explain the singulairty of the retail market in China. More information about that could be inserted and also included in the conclusive remarks

-

Author Response

Response to Reviewer 3 Comments

Point 1: The relation between buyer power and sustainability is always recalled but none even explicitly explained in the introductory part. Is it only refers to an economic sustainability, or the author want to refer to environmental or social sustainability also? Reading the article it is evident that we refer mainly on the first idea of sustainability, but in the introductory part maybe the other two could be taken into account, specifying only at the end that the author will be focus his analysis on economic/market sustainability.

Response 1: Your suggestion is very important. According to your suggestion, we have already distinguished the environmental, social and economic sustainability in the paper, and show that our sustainability mainly refers to economic sustainability.

Point 2: It is clear that buyer power could be unsustainable for the suppliers (reducing margins or putting some of them outside of the market) or customers (downloading on them the cost of the retailer buyer power) but it also could push suppliers to reduce the quality of their productive process or their product impacting negatively on the environment or the health of consumers. Despite the paper face the effect on price and competition, some consideration about the different form of impact on sustainability generated by the buyer power of mass retailers has to be mentioned and explained, mainly in the introduction or the literature review paragraph.

Response 2: With reference to Biely et al. (2018), we have added some other forms of market power that influence sustainability in the introduction and literature review, not only by influencing price and competition, but also by influencing innovation, food. security and corporate reforms, etc.

Point 3: It less explained the socio-economical peculiarity of the Chinese retail markets. How can we explain its characteristics and peculiarities? What specific historical, normative or institutional environment could explain the singularity of the retail market in China? More information about that could be inserted and also included in the conclusive remarks

Response 3: The characteristics of the Chinese retail market are caused by a variety of reasons. In the past, due to the influence of the planned economy, China’s retail industry was dominated by traditional, single department stores. When China received the admission of WTO, Carrefour, Wal-Mart, Tesco and other well-known large-scale retail organizations have entered China. It has transformed China’s retail market into a form of coexistence of various formats such as large supermarkets, shopping malls, discount stores, specialty stores, hypermarkets, department stores, and convenience stores. In addition, the entry of large foreign retailers has also affected the structure of China’s retail market. The huge scale of foreign large-scale retailers themselves and the merger and acquisitions have made the concentration of retail market in China continue to increase. Large-scale retailers such as Wal-Mart, Best Buy, Tesco, Hualian, and China Resources Vanguard have continuously improved their market position in the industrial chain, and their purchasing power has been increasing relative to upstream suppliers.

However, with the rapid development of Internet technology, emerging online retail companies have emerged, forming an online retail giant represented by Tmall, Taobao, Jingdong Mall and Dangdang. Online retailers have become another major feature in China's retail market. The sales volume of online retailers increases year by year, the market share is getting bigger and bigger, and the market share of physical retailers is squeezed out, making the Chinese retail market not as high as the developed countries in Europe and America, so there is no serious polarization in the Chinese retail market. According to the statistics of the China Chain Store & Franchise Association (CCFA), the sales volume of China’s top 100 chain companies reached 2.2 trillion in 2017, accounting for 6.0% of the total retail sales of consumer goods, compared with 11% in 2010.

In addition, in the conclusion section, we have added some inspirations from the conclusions. How to regulate the buyer power of large retail organizations? This issue has become a concern of the anti-monopoly of various countries. However, there are no specific provisions in China’s Anti-monopoly Law to regulate the buyer’s power. This may be an institutional feature in the Chinese retail market.

Round 2

Reviewer 1 Report

I understand what you would like to do. However, I still have reservations on the paper.

In the reply letter, you wrote "We assume that a powerful national retailer can prioritize the retail price, while local retailer can only follow the retail price. This change in the order of the game is similar to the change from Bertrand to Stackelberg, ..."
I understand your intention. However, I still have a serious question on your assumption. Do you know the quite famous fact that the price follower has profitability advantage over the price leader under the price competition model using your demand system when the marginal costs are the same? That is, do you know the so-called "second-mover advantage" under price competition? Following the fact, I have a doubt whether the timing structure captures your intention to explain the scenario in which the "powerful" retailer also has an advantage over its rival in the retail level. Although I buy your intention, I would like to know how do you think about the quite famous fact? Do you think that the timing structure suitably captures your intention? If you really think so, you carefully explain why you think so, by carefully taking into account the effect of "second-mover advantage".

Although you clearly mention the difference between your paper and Erutku (2005) in the reply, why don't you explain the fact in the main text? This is a quite bad manner. You should carefully write the key difference between them in the main text.

Reviewer 2 Report

The new version of the manuscript reads well henceforth the paper can be accepted for publication.

Author Response

Point 1: The new version of the manuscript reads well henceforth the paper can be accepted for publication.

Response 1: Thank you very much for your recognition, we will continue to improve our manuscript.

Reviewer 3 Report

I think that the improvements are sufficient and the article is ready for pubblication. 

Author Response

Point 1: I think that the improvements are sufficient and the article is ready for publication. 

Response 1: Thank you very much for your recognition, we will continue to improve our manuscript.

Round 3

Reviewer 1 Report

Thank you very much for your excellent effort to persuade me.

I am almost satisfied with the revision.

I have only one minor concern. Could you explain where the difference between your paper and Erutku (2005) (paper number 39) is explained? Although I tried to find the statement, I could not find it.  Could you clearly point out where?

Author Response

Thank you very much for your recognition.

Maybe you are reviewing the PDF file on the submission system, which was uploaded by the editor in round 2. I also saw that the PDF file did not reflect this modification. Please check the Word file.

I have shown the main difference between us and Erutku (2005) in the Word file I uploaded. The details as follows:

line 315-316: The essential difference between the paper and Erutku is the three-stage game, Erutku used a two-stage game to allow R1 and R2 sets retail price simultaneously [39].

line 386-388: This is another difference between this paper and Erutku, we analyzed the vertical direct effect and horizontal indirect effect of buyer power, while Erutku only considered the vertical effect [39].

line 389-391: In order to eliminate the influence of the horizontal effect on the wholesale price decision, consider an ’assumption scenario’ where the manufacturer and retailer conduct a two-stage game, this method is the same as Erutku [39].

line 481-482: This is another major difference between this paper and Erutku, who does not consider the impact of retailer competition on the price effect of buyer power [39].